

# Seroprevalence and silent infection rate during SARS-CoV-2 pandemic among children and adolescents in Western Pomerania: a multicenter, cross-sectional study—the COVIDKID study

Marcus Vollmer[1,*], Daniela Kuthning[2,*], Jana Gramenz[2], Arevik Scholz[2], Robin Michael[2], Nico Wittmann[2,3], Udo Gesser[4], Christian Niesytto[5], Antje Vogler[6], Vanda Tuxhorn[7], Ute Lenschow[8], Anja Lange[2], Kristina Linnemann[9], Holger Lode[10], Lars Kaderali[1] and Almut Meyer-Bahlburg[2,3]

[1] Institute of Bioinformatics, University Medicine Greifswald, Greifswald, Germany
[2] Pediatric Rheumatology, Department Pediatric and Adolescent Medicine, University Medicine Greifswald, Greifswald, Germany
[3] German Center for Child and Adolescent Health (DZKJ). Partner site Greifswald/Rostock, Greifswald, Germany
[4] Children's Hospital, Sana Hospital Rügen, Bergen, Germany
[5] Department of Pediatrics, AMEOS Hospital Anklam, Anklam, Germany
[6] Department of Pediatrics, Asklepios Clinic Pasewalk, Pasewalk, Germany
[7] Department of Pediatrics, Kreiskrankenhaus Demmin, Demmin, Germany
[8] Department of Pediatrics, Helios Hanseklinikum Stralsund, Stralsund, Germany
[9] Pediatrics Outpatient Clinic, Greifswald, Germany
[10] Pediatric Hematology and Oncology, Department Pediatric and Adolescent Medicine, University Medicine Greifswald, Greifswald, Germany
[*] These authors contributed equally to this work.

Corresponding author
Almut Meyer-Bahlburg, almut.meyer-bahlburg@med.uni-greifswald.de

## ABSTRACT

**Background.** Limited data on SARS-CoV-2 seroprevalence in rural areas of northern Germany necessitate comprehensive cohort studies. We aimed to evaluate the seroprevalence, silent infection (SI) rates and risk factors for infections among children and adolescents in Western Pomerania from December 2020 to August 2022.

**Methods.** In this cross-sectional study, serum or plasma samples from children and adolescents (6 months to 17 years) were collected during routine blood draw. SARS-CoV-2 specific antibodies (S1 and nucleocapsid) and their neutralizing capacity were analyzed using commercially available enzyme-linked immunosorbent and neutralization assays. Socio-demographic data and information about SARS-CoV-2 infection or vaccination were obtained. Multivariable logistic regression was used to identify independent risk factors for SARS-CoV-2 infections and SI.

**Results.** A total of 1,131 blood samples were included into the study. Overall, SARS-CoV-2 seroprevalence was 25.1%, strongly influenced by the pandemic course, predominant virus variants, age and approval of vaccination. SI rate was 5.4% (95%-CI [3.7%–6.8%]) among unvaccinated and undiagnosed children over the entire study period with highest rates among adolescents. Main risk factor despite the time at risk for silent infections was an infected household member (Odds ratio = 9.88, 95%-CI [4.23–22.9], $p < 0.001$). Factors associated with overall infections (known and silent)

also include the infection of a household member (Odds ratio = 17.8, 95%-CI [10.7–29.6], $p < 0.001$).

**Conclusions**. We believe that the introduction of governmental measures and systematic test strategies in schools strongly impacted on the SI rate, as we suspect that asymptomatic cases have already been identified, resulting in surprisingly low SI identified in our study.

# INTRODUCTION

Since the first description of severe acute respiratory syndrome coronavirus type 2 (SARS-CoV-2) in December 2019 (*Zhu et al., 2020*) and its subsequent global expansion resulting in a pandemic (*World Health Organization, 2024*), the new coronavirus causing respiratory coronavirus disease 2019 (COVID-19) was responsible for 32,145,157 cases in Germany and 582,669 cases in the federal state of Mecklenburg-Western Pomerania (Mecklenburg-Vorpommern, MV) by the end of August 2022 (*Robert Koch Institute, 2022*). The pandemic was characterized by emergence of several variants, each defined by altered levels of infectivity, symptom presentation and immune escape (*Whitaker et al., 2022*; *Sumner et al., 2023*).

To fight the pandemic, several preventive measures were implemented. Together with the introduction of vaccination against SARS-CoV-2 for adults by the end of 2020 and subsequently for children from May 2021 (*European Medicines Agency, 2024*) they impacted the course of the pandemic. In contrast to adults, children and adolescents are often only mildly affected by acute SARS-CoV-2 infection, with only mild or no symptoms, resulting in significant under-ascertainment rate. This underlines the importance of seroprevalence studies.

In addition, infection rates in densely populated urban regions may differ from those in sparsely populated rural areas, with several studies demonstrating different results (*Kleynhans et al., 2021*; *Cantu et al., 2024*; *Anzalone et al., 2023*; *Uschner et al., 2022*; *Czerwiński et al., 2023*; *Bignami-Van Assche et al., 2024*). Seroprevalence among children for urban regions of Germany is available (*Hippich et al., 2021*; *Kirsten et al., 2022*; *Leone et al., 2022*; *Sorg et al., 2022*; *Wachter et al., 2022*; *Engels et al., 2023*), but there is less information for more rural areas like MV. Western Pomerania is covered by three districts, which belong to the 6% least populated districts in Germany (each with 60/70/47 inhabitants per km$^2$) (*StatistischesBundesamt, 2023*).

In our present study, we aimed to analyze temporal changes of seroprevalence for SARS-CoV-2 in Western Pomerania, estimate the rate of silent infections (SI) and identify associated risk factors. With this, our study sheds light on the dynamic changes in seroprevalence and rate of SI in children and adolescents during SARS-CoV-2 pandemic,

most importantly with the emergence of Omicron, as many studies concentrated on earlier phases of the pandemic.

## MATERIALS & METHODS

### Cohort selection

For this multicenter, cross-sectional study, participants (6 months to 17 years) with permanent residence in Western Pomerania were recruited in six participating pediatric hospitals and two outpatient practices. The indication for participation in the study was routine blood sampling at the participating clinics. Our study itself was not the reason for the blood sampling, but rather general indications, such as the monitoring of laboratory parameters in chronically ill children, the planned blood sampling for further diagnostics in children with certain pathological symptoms (*e.g.*, arthralgia) or for further diagnostics in acutely ill children and adolescents. Participation included written informed consent, collection of serum or plasma during routine blood draw and completion of a questionnaire. Three patients with known primary immunodeficiencies or under immunoglobulin replacement therapy were excluded from the study. Repeated participation after six months was allowed as exposure to the virus, life circumstances, and serology may have changed in the interim. The children were categorized in three distinct age groups: children under 5 years of age, between 5 and 11 years old (school children), and between 12 and 17 years old (adolescents).

The study was registered in the German Clinical Trials Register on March 9, 2021 (Trial ID: DRKS00024635).

### Outcomes

SARS-CoV-2 specific antibodies in serum or plasma were analyzed by enzyme-linked immunosorbent assays (ELISA), using commercially available kits (EUROIMMUN AG, Lübeck, Germany) against the S1 domain (IgG-S1 and IgA-S1) and the nucleocapsid antigen (NCP-IgG) according to the manufacturer's instructions. Samples were considered positive at a ratio $\geq 1.1$, as recommended by the manufacturer. (Borderline) positive sera and samples from SARS-CoV-2 infected participants were tested for total neutralizing antibodies (NAbs) with SARS-CoV-2 Surrogate Virus Neutralization Test cPass (GenScript Biotech, Piscataway, NJ, USA). Data from this semi-quantitative test are presented as percentage of inhibition with $\geq 30\%$ inhibition capacity classified as positive.

Seropositivity was asserted when one of the following conditions was true: (1) positive IgG-S1; (2) borderline IgG-S1 and at least one additional positive test; (3) IgG-S1 negative, but at least 2 other positive tests. For the assessment of risk factors for both overall infection (OI) and SI, all vaccinated participants and those with incomplete information about their vaccination or infection history were excluded. SI was defined as seropositivity in participants without a history of SARS-CoV-2 infection or vaccination reported by questionnaire. OI was defined as either samples tested seropositive or participants declared an infection anamnestically.

## Statistical analysis

The analysis of the study includes the use of external data (see Methods S1). Official register data was used to check the representativeness of age, sex, and vaccination rate of our children's sample. Centered 7-day averages of daily case numbers in the three districts were used to compute the incidence of daily infections per 1,000 children in the respective age groups. Turning dates of the new dominating variants were interpolated from freely available data from the CoMV-Gen study center, which reported weekly fractions of challenging variants (interpolated fraction >50% for respective variant) determined from variant-polymerase chain reaction (PCR) or whole genome sequencing (*Kohler et al., 2023*; *CoMV-Gen, 2023a*; *CoMV-Gen, 2023b*; *CoMV-Gen, 2023c*; *CoMV-Gen, 2023d*).

Three study periods were defined based on dominating variant: Alpha, Delta and Omicron waves. Time course analysis was done using locally weighted scatterplot smoothing (LOESS). SI were compared among the age groups for each study period by Fisher's exact test. To evaluate associating factors, multivariable logistic regression were performed for OI and SI. Since this requires complete data, we created 10 imputed data frames using random forest imputation (*Shah et al., 2014*). We applied four-fold cross-validated elastic net with a fair mix of L1- and L2-regularization (alpha =0.5) onto the imputed data frames to reduce model size for better generalizability (*Tay, Narasimhan & Hastie, 2023*). The extracted variables at a specific lambda value (one standard error from optimal lambda) entered the final model formula. In the following, we trained the regression coefficients on each imputed data frame and pooled the estimations to increase reliability. As blood were sampled at different dates and stages of the pandemic, the time exposed to an infectious environment and the severity of infectiousness was taken into account when modeling OI and SI. We therefore have cumulated the incidence numbers reported since the start of the pandemic till blood sampling date of the respective age groups and districts. The infection risk variable was defined as the resulting cumulative numbers normalized by the number of children in each age group of the three districts to allow a better comparison of the estimated odds ratios (OR). We performed Type-III likelihood ratio tests and computed OR with 95% confidence intervals from the regression models. Multicollinearity was checked using variance inflation factors (VIF) with default cutoffs: VIF>5 for concern, VIF>10 for serious collinearity (*James et al., 2021*).

All statistical tests were conducted two-sided with R version 4.4.1 (*R Core Team, 2024*). The significance level was set a priori at 5%.

## Ethics approval

This study was performed in line with the principles of the Declaration of Helsinki. Ethics approval was obtained from the Ethics Committee of University Medicine Greifswald (BB188/20 and BB188/20a).

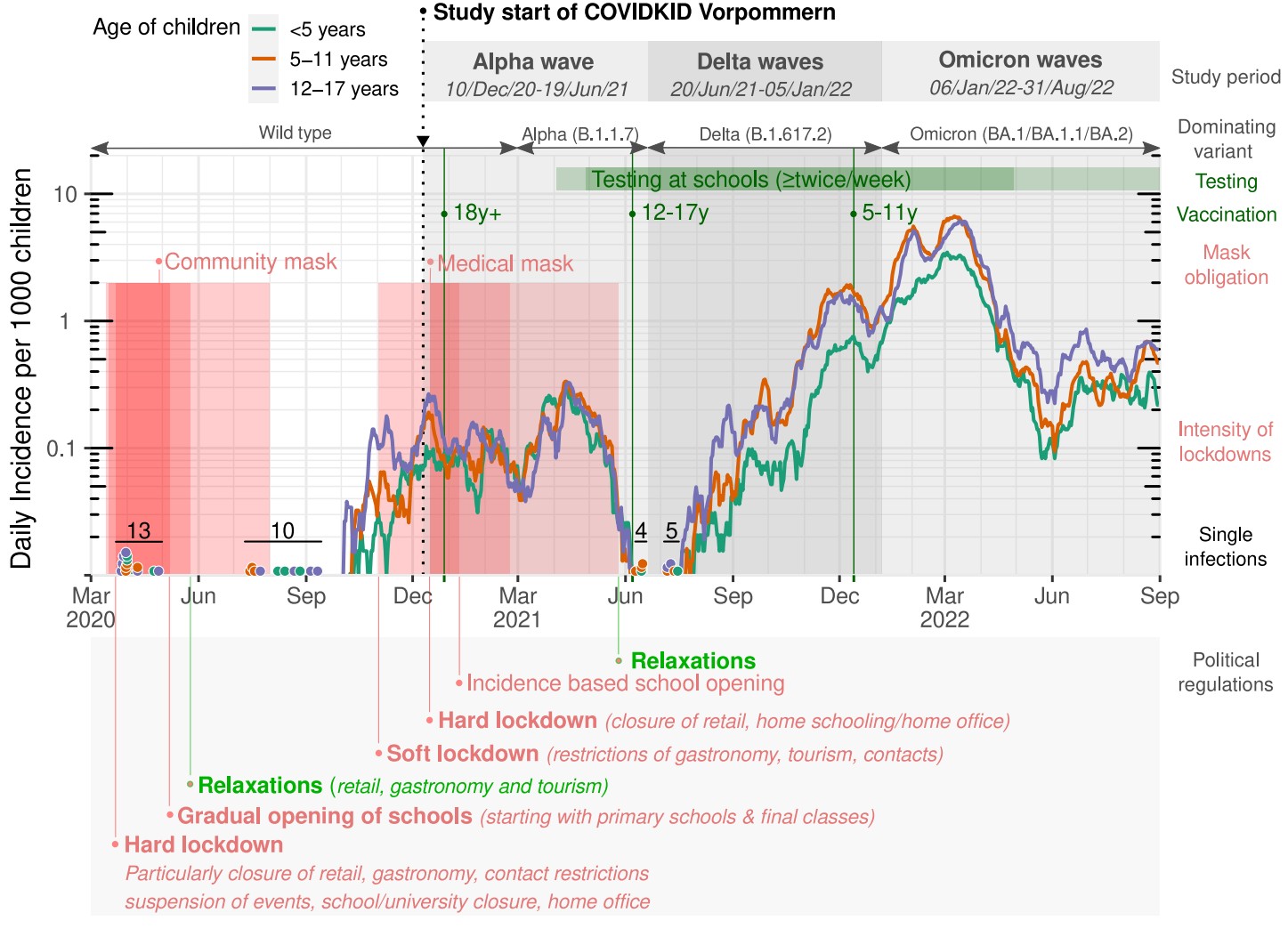

**Figure 1** **Infection rates in children and adolescents in the study region of Western Pomerania, Germany.** Regional incidences per 1,000 children for each age group printed as 7-day average, with single infections shown as dots. Dominating variants were calculated from variant PCR (change points were defined at 50% domination). Study periods of the COVIDKID study are defined at these change points. Start of vaccination campaigns for adults and children are indicated by green bars, mandatory wearing of masks is indicated in red. Red areas showing lockdown measures depending on restriction intensity and relaxations labeled at major events.

## RESULTS

### Course of virus variants, political decisions, vaccination and SARS-CoV-2 infection in the study region

The course of the pandemic was evaluated for the study area as defined in the methods section. Normalized incidences of SARS-CoV-2 infection, approval of vaccination starts in different age groups, mask obligations and preventive measures implemented by the federal state government of MV are summarized in Fig. 1. The amount of available data in the particular variables can be read from column N of Table 1.

**Table 1** Characteristics of the study population stratified by age.

| Variable | N | Included children in Western Pomerania, Germany | | |
|---|---|---|---|---|
| | | *Children under 5* Age <5 years N = 277 | *School children* Age 5–11 years N = 400 | *Adolescents* Age 12–17 years N = 454 |
| Sex, n (%) | | | | |
| Male | | 144 (52.0) | 215 (53.8) | 203 (44.7) |
| Female | | 133 (48.0) | 185 (46.2) | 251 (55.3) |
| Sampling at, n (%) | 1,131 | | | |
| Alpha wave (10/Dec/2020-19/Jun/2021) | | 87 (31.4) | 167 (41.8) | 214 (47.1) |
| Delta waves (20/Jun/2021-05/Jan/2022) | | 103 (37.2) | 104 (26.0) | 103 (22.7) |
| Omicron waves (06/Jan/2022-31/Aug/2022) | | 87 (31.4) | 129 (32.2) | 137 (30.2) |
| Sample taken after approval of vaccination, n/N (%) | 1,131 | 0/277 (0.0%) | 139/400 (34.8%) | 254/454 (55.9%) |
| Self-disclosure about vaccination and infection, n (%) | 1,102 | | | |
| Vaccinated and infected | | 1 (0.4) | 6 (1.5) | 25 (5.6) |
| Vaccinated only | | 1 (0.4) | 15 (3.8) | 86 (19.4) |
| Infected only | | 44 (16.5) | 77 (19.6) | 41 (9.3) |
| Neither vaccinated nor infected | | 220 (82.7) | 295 (75.1) | 291 (65.7) |
| IgG-S1, Median (IQR) | 1,131 | 0.13 (0.08, 0.49) | 0.14 (0.09, 0.48) | 0.17 (0.10, 3.93) |
| IgA-S1, Median (IQR) | 1,130 | 0.12 (0.07, 0.34) | 0.10 (0.06, 0.28) | 0.09 (0.06, 0.23) |
| IgG-NCP, Median (IQR) | 1,129 | 0.20 (0.13, 0.38) | 0.23 (0.16, 0.43) | 0.34 (0.19, 1.54) |
| NAbs (%), Median (IQR) | 376[a] | 21 (6, 87) | 35 (7, 91) | 94 (32, 97) |
| Seropositive, n/N (%) | 1,131 | 53/277 (19.1%) | 82/400 (20.5%) | 149/454 (32.8%) |
| Overall infection in unvaccinated children, n (%) | 968 | | | |
| Not infected | | 202 (76.5) | 283 (76.1) | 280 (84.3) |
| Infected (Self-disclosed or SI) | | 62 (23.5) | 89 (23.9) | 52 (15.7) |
| Day care facility, n (%) | 1,042 | | | |
| Not in a day care facility | | 36 (14.2) | 7 (1.9) | 40 (9.6) |
| Nursery/Nanny | | 114 (44.9) | 1 (0.3) | 0 (0.0) |
| Kindergarten | | 101 (39.8) | 71 (19.1) | 1 (0.2) |
| School with after-school care | | 1 (0.4) | 71 (19.1) | 3 (0.7) |
| School without after-school care | | 2 (0.8) | 222 (59.7) | 372 (89.4) |
| Medication intake at sampling, n/N (%) | 1,079 | 165/258 (64.0%) | 168/380 (44.2%) | 231/441 (52.4%) |
| Chronic diseases, n (%) | 1,077 | | | |
| No chronic diseases | | 219 (84.2) | 268 (70.3) | 283 (64.9) |
| Chronic diseases except respiratory | | 29 (11.2) | 79 (20.7) | 127 (29.1) |
| Chronic diseases including respiratory | | 12 (4.6) | 34 (8.9) | 26 (6.0) |
| Smoking parent, n/N (%) | 1,091 | 117/262 (44.7%) | 164/388 (42.3%) | 202/441 (45.8%) |
| Child is smoking, n/N (%) | 1,095 | 3/263 (1.1) | 9/389 (2.3) | 47/443 (10.6) |
| Self-disclosure about infection, n (%) | 1,081 | | | |
| Neither parents nor child | | 186 (71.8) | 285 (74.0) | 348 (79.6) |
| Only a parent | | 33 (12.7) | 20 (5.2) | 27 (6.2) |
| Only child | | 5 (1.9) | 11 (2.9) | 18 (4.1) |
| Parent and child | | 35 (13.5) | 69 (17.9) | 44 (10.1) |

**Table 1** (*continued*)

| Variable | N | Included children in Western Pomerania, Germany | | |
|---|---|---|---|---|
| | | *Children under 5* Age <5 years N = 277 | *School children* Age 5–11 years N = 400 | *Adolescents* Age 12–17 years N = 454 |
| Household member works in the healthcare system with patient contact (including nursing home, nursing service and others), n/N (%) | 1,080 | 84/262 (32.1) | 123/379 (32.5) | 135/439 (30.8) |
| Highest level of education of the household, n (%) | 860 | | | |
| no high school diploma (yet) | | 6 (2.6) | 11 (3.7) | 26 (7.8) |
| Secondary school ("Hauptschule") | | 13 (5.7) | 21 (7.0) | 22 (6.6) |
| Junior high school ("Mittlere Reife") | | 88 (38.8) | 119 (39.5) | 143 (43.1) |
| University entrance qualification ("Hochschulreife") | | 48 (21.1) | 63 (20.9) | 77 (23.2) |
| Academic degree ("Hochschulabschluss") | | 72 (31.7) | 87 (28.9) | 64 (19.3) |
| Number of household members, *n* (%) | 1,063 | | | |
| 1 or 2 | | 10 (3.9) | 23 (6.1) | 42 (9.8) |
| 3 | | 75 (29.1) | 97 (25.7) | 122 (28.5) |
| 4 | | 120 (46.5) | 163 (43.2) | 149 (34.8) |
| 5 | | 36 (14.0) | 67 (17.8) | 59 (13.8) |
| 6 or more | | 17 (6.6) | 27 (7.2) | 56 (13.1) |
| Household composition, n/N (%) | | | | |
| Small child (<5 years)[b] | 748 | 61/177 (34.5%) | 68/271 (25.1%) | 39/300 (13.0%) |
| Senior (≥60 years)[b] | 748 | 86/177 (48.6%) | 122/271 (45.0%) | 118/300 (39.3%) |
| Animals | 1,029 | 123/248 (49.6%) | 221/359 (61.6%) | 254/422 (60.2%) |
| Traveled beyond the district during the past 6 months, n/N (%) | 1,096 | 112/263 (42.6%) | 183/389 (47.0%) | 185/444 (41.7%) |
| Burdens of the respondent parent considered to be the most serious, n/N (%) | 886 | | | |
| Occupational insecurity | | 53/229 (23.1%) | 64/311 (20.6%) | 66/346 (19.1%) |
| Fear of their own illness | | 71/229 (31.0%) | 71/311 (22.8%) | 79/346 (22.8%) |
| Fear of illness in close relatives | | 106/229 (46.3%) | 130/311 (41.8%) | 151/346 (43.6%) |

**Notes.**
[a]Only positive or borderline positive sera as well as samples from patients reporting a history of SARS-CoV-2 infection were tested for total neutralizing antibodies (NAbs).
[b]Belongs to the household or is one of the very close contacts of the household community.

## Characteristics of study population in Western Pomerania

In total, 1,131 samples from 1,093 children and adolescents were included. Median age at sample collection was 10 years (quartile range 5 to 14 years). Sex distribution across the three age groups was slightly, but significantly different (female: 48.0%, 46.2% and 55.3%, $p = 0.02$). Overall, the pediatric population of the study region is well represented, as the observed participant numbers follow an ideal representative age distribution and vaccination rates estimated from the study sample follows official numbers from RKI monitoring (see Figs. S1–S3). 1-year old children might be overrepresented in the the study sample (95 in the sample, 57 expected).

As the recruitment was hospital-based and involved recruitment in several special out-patient clinics, 28.5% of participants have at least one pre-existing condition. Of note, 17.9% of patients reported at least one known SARS-CoV-2 infection, 12.0% of participants

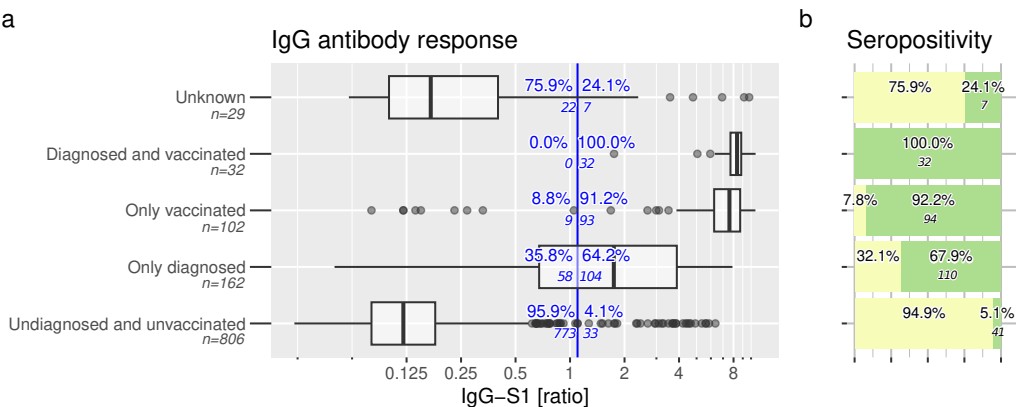

**Figure 2** **Anti-SARS-CoV-2 IgG response and seropositivity.** Anti-SARS-CoV-2-antibodies are evident in children and adolescents. (A) Anti-S1 IgG response and (B) seropositivity of all study participants at time of study inclusion, contingent upon self-disclosed SARS-CoV-2 vaccination and infection status. Vertical blue line indicates manufacturer's recommended cut-off (ratio IgG-S1 ≥ 1.1). Seropositivity (green boxes in B) was inferred based on defined rules. Samples with missing information about infection and vaccination status were categorized as "unknown".

disclosed at least one vaccination. 21.1% of all children and adolescents reported to had contact to at least one SARS-CoV-2 infected household member. Participants characteristics and serological test results stratified by age are summarized in Table 1.

## Estimation of seroprevalence among children and adolescents during the pandemic in Pomerania

Figure 2A shows the primary outcome of IgG-S1 antibodies. The highest rate of IgG-S1-positivity was achieved in vaccinated participants with or without prior SARS-CoV-2 infection (32/32 =100% and 93/102 =91.2% respectively). In addition, 64.2% (104/162) of unvaccinated but infected participants were positive for IgG-S1. In the group of SARS-CoV-2 undiagnosed and unvaccinated participants, 4.1% were positive for IgG-S1 (33/806).

Overall seropositivity rate among these groups is shown in Fig. 2B and differs slightly from IgG-S1 positivity rates. Detailed information of tested participants is displayed in Table S1 and Figs. S4–S5.

For all age groups, seroprevalence increases in late 2021 (Fig. 3). In adolescents, a rapid increase of "vaccinated or recovered" participants was found with the estimated seroprevalence reaching comparable rates of approximately 90% at the end of the study. In school children, we see a strong deviation between both rates with approximately 90% "vaccinated or recovered" but an estimated seroprevalence of only about 60%. In children under 5, seroprevalence rates resemble those of school children despite slightly higher rates until mid-2021 and a lower rate of anamnestically recovered children. Of note, estimated seroprevalence and the rate of "vaccinated or recovered" children under 5 are diverging, especially during spring/summer 2021 and from May 2022 with emerging Omicron. At the end of the study, nearly 70% of this group were seropositive. In summary, at the end of

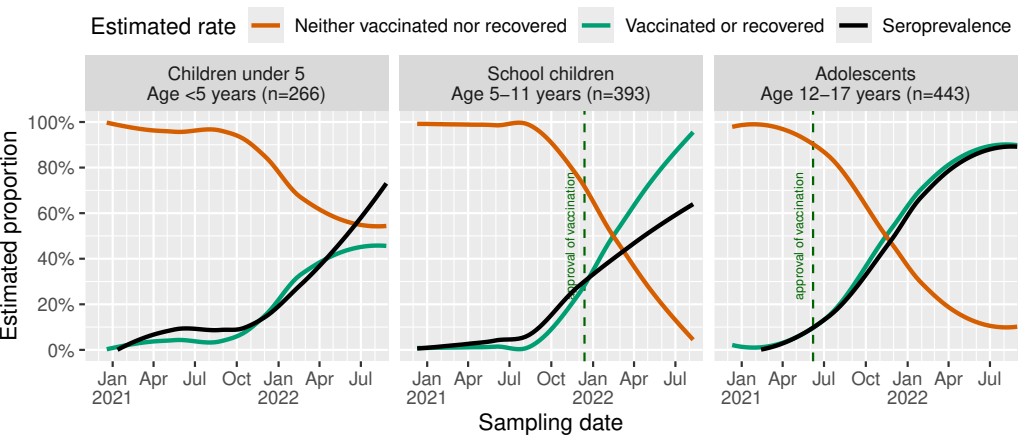

**Figure 3** **Estimated rates of recovered or SARS-CoV-2-vaccinated children and seroprevalence.** Seroprevalences across age groups were estimated using LOESS. Dashed lines shows the official vaccination start in the study region depending on the respective age group.

**Table 2** **Silent infection rates among undiagnosed and unvaccinated children and adolescents in Western Pomerania, stratified based on the time of study enrollment.**

| Study period | Children under 5 Age <5 years | School children Age 5–11 years | Adolescents Age 12–17 years | p-value[a] |
|---|---|---|---|---|
| Alpha wave (10/Dec/20-19/Jun/21), n/N (%) | 3/79 (3.8%) | 4/163 (2.5%) | 2/204 (1.0%) | 0.23 |
| Delta waves (20/Jun/21-05/Jan/22), n/N (%) | 4/89 (4.5%) | 7/93 (7.5%) | 2/62 (3.2%) | 0.55 |
| Omicron waves (06/Jan/22-31/Aug/22), n/N (%) | 11/52 (21.2%) | 1/39 (2.6%) | 7/25 (28.0%) | 0.006 |

**Notes.**
Notation: Number of silent infections/Total number of naive children or adolescents (silent infection rate in %).
[a] Fisher's exact test.

August 2022, children and adolescents showed seroprevalence of 60% to 90%. During the first wave, we noticed slightly higher rates among children under 5. For subsequent Delta and Omicron waves, it reverses to the opposite with increasing rates with age.

### Estimation of silent infection rate

In the presumptive negative samples, 41 undetected infections were confirmed, which corresponds to an SI rate of 5.4% (41/806, 95% Clopper–Pearson confidence interval 3.7% to 6.8%, see Fig. 2B). The distribution of these identified infections across age groups and major variants are summarized in Table 2. While the SI rates during Alpha dominance ranged between 1.0% to 3.8%, with no significant association with age, we found an increase in SI rate in all age groups, with significantly higher rates during Omicron dominance in children under 5 (11/52 = 21.2%) and in adolescents (7/25 = 28.0%) compared to school children (1/39 = 2.6%, Fisher's exact test $p = 0.006$). However, the absolute number of undiagnosed and unvaccinated adolescents remarkably drops during Omicron domination.

Next, we analyzed IgG-S1 seroprevalence within the three groups in respect to date of study inclusion. Under strong restrictions (Fig. 4, marked in red shading), only 10 infections were diagnosed (upper red triangles), particularly from March 2021 on, and 7 SI were
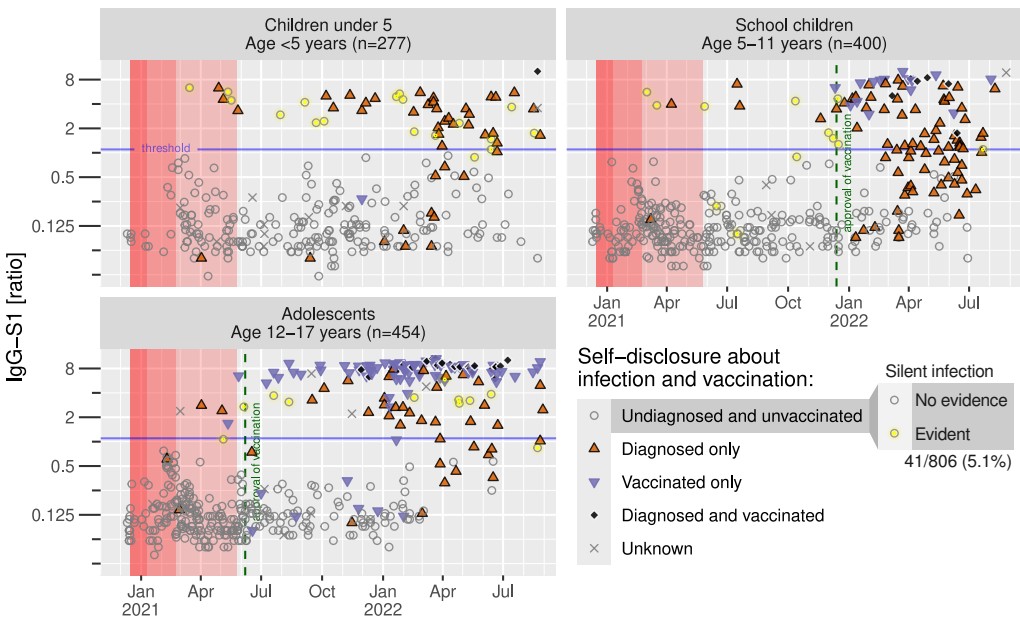

**Figure 4** **Anti-SARS-CoV-2-antibodies in time course.** Individual anti-S1 antibodies of recruited participants related to their anamnestic infection and vaccination status. Dashed lines indicating vaccination approvals in respective age groups, red areas indicating infection control measures in Western Pomerania, depending on harshness of restrictions.

confirmed across all age groups until June 2021. The number of vaccinated participants (lower blue-magenta triangles) rapidly increased from June 2021 in adolescents and November 2021 in school children. Bulk infections were identified from November 2021 in all age groups. It is noteworthy that most school children were aware of their infection after the start of the vaccination campaign in January 2022 (72 of 138 reported an infection). Only three silent infections were identified among the remaining 46 undiagnosed and unvaccinated children.

## Evaluation of potential risk factors for OI and SI

Table 3 shows the results of the multivariable logistic regression for OI and SI. OI was significantly associated with infection risk (OR $=1.00$, 95%-CI [0.96–1.05] [children under 5], 1.02, 95%-CI [0.98–1.05] [school children], 1.05, 95%-CI [1.01–1.09] [adolescents] for each percent increase in the case numbers in the respective age group). Moreover, the dominating variant at the time of study enrollment is significantly associated with OI, with significantly higher OR during Delta and Omicron waves compared to the Alpha wave (OR $=2.52$, 95%-CI [1.33–4.77], $p = 0.004$ and OR $=6.57$, 95%-CI [2.41–17.9], $p < 0.001$, respectively). Our results demonstrate the strongest association for OI and SI when a household member was diagnosed with SARS-CoV-2 infection (OR $=17.8$, 95%-CI [10.7–29.6], $p < 0.001$ and OR $=9.88$, 95%-CI [4.23–22.9], $p < 0.001$, respectively). All other socio-demographic characteristics considered did not make it into the regression equation.

**Table 3   Results of logistic regressions for associations with overall and silent infections.**

| Characteristic | Overall infections (OI) among unvaccinated ($n = 968$) | | Silent infections (SI) among unvaccinated and uninfected ($n = 806$) | |
|---|---|---|---|---|
| | OR (95% CI)[a] | *p*-value | OR (95% CI)[a] | *p*-value |
| Infection risk[b] in interaction with age [OR per 1% increase of infected] | | | | |
| Children under 5 (age: <5 years) | 1.00 (0.96 to 1.05) | 0.88 | 1.05 (1.01 to 1.10) | 0.016 |
| School children (age: 5-11 years) | 1.02 (0.98 to 1.05) | 0.35 | 0.95 (0.91 to 1.0) | 0.045 |
| Adolescents (age: 12-17 years) | 1.05 (1.01 to 1.09) | 0.011 | 1.03 (0.98 to 1.08) | 0.216 |
| Dominating variant | | | | |
| During Alpha wave (10/12/2020-19/06/2021) | 1.00 (Reference) | | | |
| During Delta waves (20/06/2021-05/01/2022) | 2.52 (1.33 to 4.77) | 0.004 | | |
| During Omicron waves (06/01/2022-31/08/2022) | 6.57 (2.41 to 17.9) | <0.001 | | |
| At least one household member was infected | 17.8 (10.7 to 29.6) | <0.001 | 9.88 (4.23 to 22.9) | <0.001 |

Notes.
[a]OR, multivariable adjusted Odds Ratio, CI, Confidence Interval of the OR.
[b]Infection risk is increasing with the time at risk and defined by the cumulative cases reported to LAGuS of each age group of the three study districts. Cumulative case numbers were divided by the total population of children in the respective age group of all three districts to allow meaningful interpretation of OR.

## Power of performed serological tests

We noticed an increasing number of samples found to be positive for IgG-S1 but negative for all other serological tests, most notably neutralizing antibodies testing by cPass (Fig. 5A). When comparing IgG-S1 positive and negative samples from diagnosed participants before and after 6 January 2022, our data demonstrate a significant loss of power for NAbs and IgA-S1 with emerging Omicron variants, as only 10 of 44 IgG-S1-positive samples were also NAb-positive (see top-right of Fig. 5B) and only 15 of 46 IgG-S1-positive samples were also IgA-S1-positive. IgG-S1 and IgG-NCP suffered from decreased antibody ratios but were still able to detect antibodies during the Omicron wave to a similar extent as during previous waves. In summary, IgG-S1 performed best to identify positive samples after confirmed infection, and IgG-NCP is particularly useful to detect antibodies irrespective of SARS-CoV-2-variants and vaccination just a few weeks after infection.

# DISCUSSION

## Principal findings

In our study we aimed to investigate the seroprevalence of SARS-CoV-2 specific antibodies in children and adolescents in Western Pomerania, part of a rural federal state in northeast of Germany from mid-December 2020 to August 2022. At the beginning of the pandemic, immunity against SARS-CoV-2 and SI rates were extremely low in all age groups of our study cohort. Seroprevalence increased during the Delta and Omicron waves with vaccine approval and relaxation of hygiene measures. Household contacts with infected individuals posed highest risk for SI. The SI rate was remarkably low at the start of our study, ranging

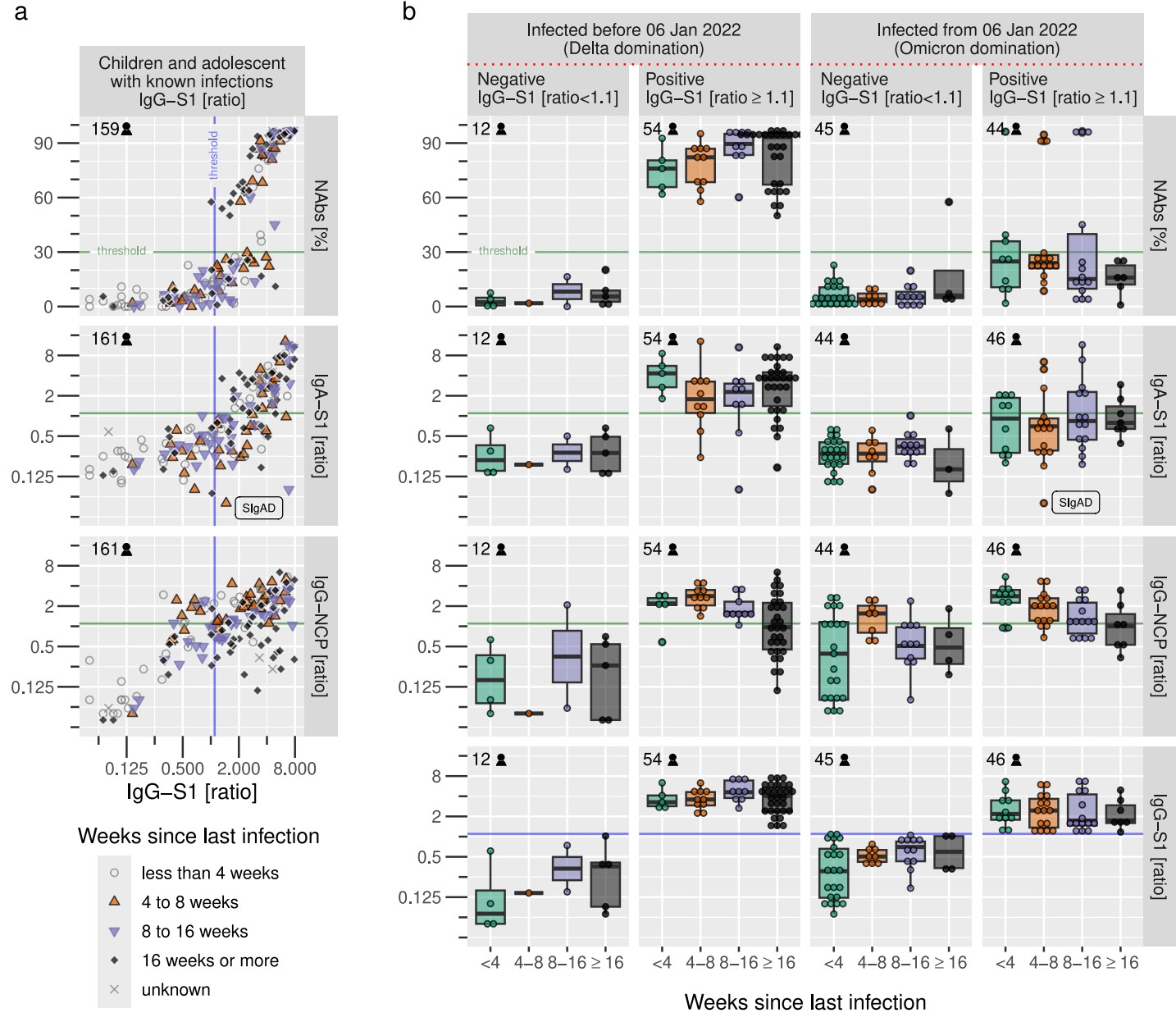

**Figure 5  Sensitivity of serological tests before and after emerging Omicron variants.** (A) Correlation of IgG-S1 ratio with other serological tests in relation to the time since self-reported infection in children and adolescents, (B) Sensitivity of serological tests in combination with the IgG-S1 ratio and the dominating variant at the time of infection. Horizontal and vertical lines indicate the manufacturer's cut-offs for the respective tests.

from 1.0% to 3.8%, and increased particularly with the rise of Omicron variants in an age-dependent manner from 2.6% to 28.0%. However, this 28% SI rate in adolescents during Omicron should not be overestimated because the majority of all adolescents had already been vaccinated or knowingly infected. There were only 25 supposedly immunologically naïve adolescents in which we were able to detect an undetected infection in 7 subjects.

## Interpretation

Initial seroprevalence rates were very low in all age groups under strict restrictions, with first infections detected in participants recruited in March 2021, and remained low under strict contact restrictions resulting in only few infections. However, seropositivity strongly increased with both the approval of vaccinations and the relaxation of contact restrictions during emerging Delta and especially Omicron waves. At the end of the study in August 2022, adolescents showed a high estimated seroprevalence of 90% whereas children under 12 years of age had a seroprevalence rate of only about 60% to 70%, suggesting a higher immunity in adolescents than in younger children. This is in line with comparable studies across Europe (*Engels et al., 2023*; *Boey et al., 2022*; *Ratcliffe et al., 2023*; *Sapronova et al., 2023*). Interestingly, our data from the first study period until June 2021 demonstrated a higher seroprevalence among very young children under 5 similar to the results from other German studies (*Leone et al., 2022*; *Sorg et al., 2022*).

The slight overrepresentation of 1-year-old children is most likely due to the fact that children of this age are hospitalised more frequently than older children and that we recruited the majority of patients in hospital. However, we do not believe this affects the overall result of the study because we stratified our analyses by age group.

In school-aged children, our data suggested the lowest immunity with insufficient rates of antibody production or detection. In contrast, seroprevalence was significantly higher than anamnestically reported infection rates in children under 5, indicating a relatively high rate of SI in this group. Other seroprevalence studies from Germany showed similar results for children in different age groups, with low seroprevalences in low-incidence settings until late 2020 and increasing rates during 2021 (*Leone et al., 2022*; *Wachter et al., 2022*; *Engels et al., 2022*; *Jank et al., 2023*). Only limited seroprevalence data are available for north-eastern Germany with the exception of the "Fr1da" study who is reporting a seroprevalence of 76.7% in children between August 2021 and June 2022 (*Ott et al., 2022*).

Our study aimed to provide additional information on seroprevalences in children of different age groups in a rural region. Several studies suggest differences in infection rates between urban and rural areas, with contrasting results, but only few studies have analysed the paediatric population: In a study from South Africa seroprevalence was lowest in children under 5 years of age in rural communities and highest in adults in urban areas (*Kleynhans et al., 2022*). In contrast, younger children in rural areas were more likely to be hospitalized for SARS-CoV-2 infections in a study from the US (*Cantu et al., 2024*). In Poland, higher seroprevalence rates were observed in rural compared to urban areas during early stages of the pandemic but this study did not specifically adressed the pediatric population (*Czerwiński et al., 2023*). A German/Italian study showed that the risk of dying from COVID-19 was lower for people living in rural areas, but only if hospitalisation was not required. In contrast, for those who were hospitalised in rural areas, the risk of dying was higher than in urban areas (*Bignami-Van Assche et al., 2024*). Overall, however, the seroprevalence rates in our study of children and adolescents from a rural area do not appear to differ significantly from others from urban regions in Germany. However, there is no direct comparison therefore the conclusion is somewhat limited.

In infectious disease epidemiology, silent infection refers to the unnoticed development of immune protection following an asymptomatic or clinically inapparent course of an infection. Under normal circumstances, this is a desirable effect, as a high rate of silent infection means that a large proportion of the population is immune to the pathogen in question. During the pandemic, however, the population was overwhelmingly SARS-CoV-2 naïve. To prevent too many people falling ill at the same time, which would have led to an overload of the healthcare system, SARS-CoV-2 infected people were to remain in quarantine. To avoid mass infections, it was therefore necessary and sensible to identify asymptomatically infected people. The risk for SI in our study was highest in adolescents but almost completely disappeared among school children beginning from December 2021. The revised testing strategy in schools may have led to early and rapid detection of infections in children. The testing strategy included regular testing in schools for students and employees at least twice a week, and was mandatory between 28 April 2021 and 29 April 2022. As a result, SARS-CoV-2 positive children were kept in quarantine to prevent transmission of the virus. In kindergarten, starting in April 2021, only symptomatic children were tested, while many day care facilities remained closed, which could explain the higher SI rate due to undertesting in this age group. Therefore, the testing strategy implemented in schools seems to have been successful in avoiding SI. The main risk factor for children and adolescents being silently infected was having an infected household member, followed by the time at risk and the dominating variant.

Consistent with other studies, our data could not demonstrate a protective association with a lower risk of infection or increased seroprevalence when health care workers are in the household community (*Ratcliffe et al., 2023*; *Almeida Carvalho et al., 2022*; *Steensels et al., 2020*). Other socio-demographic characteristics may also be associated with SI, but were considered to be of minor importance in the elastic net regression. Increased risk for children in lower educated households, as reported in previous studies, was not evident in our cohort (*Brinkmann et al., 2022*).

The risk of OI was higher among adolescents during Delta and Omicron waves and with at least one infected household member. Similar results were found in a pediatric cohort from the US (*Smith et al., 2023*).

We believe that the introduction of government policies and systematic testing strategies in schools and many public and health-related institutions has had a strong impact on the rate of SI in children, as we suspect that asymptomatic cases have already been identified, resulting in the surprisingly low rate of SI identified in our study.

During our analysis, we recognized that the serological results are subject to a change point as NAbs and IgA-S1 show a remarkable decrease in IgG-S1 positive samples after 6 January 2022. This decline may be attributed, at least in part, to the immune escape mechanisms of various SARS-CoV-2 variants.

## Limitations

There are also some limitations to our study. We had to rely on questionnaire-based reports of diagnosis and vaccination. Identified SI cannot be attributed to causing variants and time of infection. Furthermore, we found reduced sensitivity of the cPass test for neutralization

capacity, also described in other publications, leading to the need of variant-specific testing as performed elsewhere (*Springer et al., 2022*; *Kim et al., 2022*). A minor concern is cross-reactivity with other beta-coronaviruses which is still being debated (*Tamminen, Salminen & Blazevic, 2021*; *Wong et al., 2023*). As the recruitment was voluntary and hospital-based, a selection bias cannot be ruled out. In particular, the number of included blood samples are not uniformly distributed across the study period and there were more inclusions in spring 2021 and less inclusions in summer 2022, leading to a biased SI rate. However, a study from the US and a worldwide meta-analysis demonstrated similar seroprevalences for population-based, school-based and hospital-based studies (*Clarke et al., 2023*; *Naeimi et al., 2023*). Furthermore, the study sample follows the pediatric vaccination rate of the region and the age distribution is also representative. Therefore, we assume that our data allows the identification of under-ascertainment rates from late 2020 until August 2022, covering different SARS-CoV-2 variants in this well-defined region. Confounding variables can be missed in the multivariable models that may bias the estimation of independent effects. Moreover, we included a disproportionate number of patients with chronic diseases, including mainly rheumatic diseases, type 1 diabetes mellitus and respiratory diseases. The vast majority of these patients participate in life without any impairment, and severe courses of COVID-19 were not more common in these pediatric patient groups (*Sengler et al., 2021*; *Danne & Kordonouri, 2021*; *Edqvist et al., 2023*; *Dolby et al., 2023*). However, we cannot rule out the possibility that these children and adolescents were tested more frequently, which could have led to a reduced SI rate.

## CONCLUSIONS

The COVIDKID study showed a significant increase in seroprevalence in children and adolescents of all age groups, particularly with the approval and introduction of vaccination and the emergence of the Delta and Omicron waves. The highest risk factor for infection, known or silent, adjusted for time at risk and dominant variant is infection of a household member. In addition, we saw higher rates when the Delta and Omicron variants were dominant. In the study sample, the SI rate in undiagnosed and unvaccinated children and adolescents was relatively low at only 5.4%, which we believe is due to systematic testing strategies in schools.

## ACKNOWLEDGEMENTS

We thank all study participants and their families for participating in the COVIDKID study. We thank the teams of physicians and nurses in all participating clinics for their support in sample acquisition. Special thanks go to Roswitha Bruns, who died in August 2021, for supporting this study. We also thank the diagnostics department of Medical Microbiology (Head: Prof. Dr. Karsten Becker, Dr. Zimmermann, Kathrin Hollmann, Maria Schuparis) for sample analysis and the Institute of Clinical Chemistry (especially Dr. Winter) as well as IMD Greifswald for sample transport. For his expertise and support we thank Prof. Dr. Martin Groschup.

### Funding

This study was supported by University Medicine Greifswald, Germany and funded by the Ministry for Economics, Labour and Health Mecklenburg-Vorpommern (406-00000-2020/002-018). The funders had no role in study design, data collection and analysis, decision to publish, or preparation of the manuscript.

### Grant Disclosures

The following grant information was disclosed by the authors:
University Medicine Greifswald, Germany.
Ministry for Economics, Labour and Health Mecklenburg-Vorpommern: 406-00000-2020/002-018.

### Competing Interests

Almut Meyer-Bahlburg received funding for the project by the Ministry for Economics, Labour and Health Mecklenburg-Vorpommern. Lars Kaderali was a member of the Corona Expert Council advising the Federal Government of Germany.

### Author Contributions

- Marcus Vollmer analyzed the data, prepared figures and/or tables, authored or reviewed drafts of the article, and approved the final draft.
- Daniela Kuthning conceived and designed the experiments, performed the experiments, analyzed the data, prepared figures and/or tables, authored or reviewed drafts of the article, and approved the final draft.
- Jana Gramenz performed the experiments, authored or reviewed drafts of the article, and approved the final draft.
- Arevik Scholz performed the experiments, authored or reviewed drafts of the article, and approved the final draft.
- Robin Michael performed the experiments, authored or reviewed drafts of the article, and approved the final draft.
- Nico Wittmann performed the experiments, authored or reviewed drafts of the article, and approved the final draft.
- Udo Gesser performed the experiments, authored or reviewed drafts of the article, and approved the final draft.
- Christian Niesytto performed the experiments, authored or reviewed drafts of the article, and approved the final draft.
- Antje Vogler performed the experiments, authored or reviewed drafts of the article, and approved the final draft.
- Vanda Tuxhorn performed the experiments, authored or reviewed drafts of the article, and approved the final draft.
- Ute Lenschow performed the experiments, authored or reviewed drafts of the article, and approved the final draft.

- Anja Lange performed the experiments, authored or reviewed drafts of the article, and approved the final draft.
- Kristina Linnemann performed the experiments, authored or reviewed drafts of the article, and approved the final draft.
- Holger Lode performed the experiments, authored or reviewed drafts of the article, and approved the final draft.
- Lars Kaderali analyzed the data, authored or reviewed drafts of the article, and approved the final draft.
- Almut Meyer-Bahlburg conceived and designed the experiments, performed the experiments, analyzed the data, authored or reviewed drafts of the article, and approved the final draft.

## Human Ethics

The following information was supplied relating to ethical approvals (i.e., approving body and any reference numbers):

This study was performed in line with the principles of the Declaration of Helsinki. Ethics approval was obtained from the Ethics Committee of University Medicine Greifswald (BB188/20 and BB188/20a).

## Data Availability

The code for data processing and statistical analysis are available in the Supplemental Files.

## Supplemental Information

Supplemental information for this article can be found online at http://dx.doi.org/10.7717/peerj.18384#supplemental-information.

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
