# Peer review of "Seroprevalence and silent infection rate during SARS-CoV-2 pandemic among children and adolescents in Western Pomerania: a multicenter, cross-sectional study—the COVIDKID study"

_PeerJ, doi:10.7717/peerj.18384_

## Round 0.1 · original submission · Major Revisions

The reviewers agreed that your manuscript requires major revisions. Please address all of their points before resubmission, in particular you need to improve the information on representativeness or selection bias of your study participants.

·

Basic reporting

no comment

Experimental design

no comment

Validity of the findings

I am not sure if and where the underlying data are provided.

Additional comments

Representativeness: the authors only briefly describe their study cohort but not in enough detail. „…collection of serum or plasma during routine blood draw…“ I am not aware that there are any regular blood draws from children in Germany. Which were the reasons for the blood draws. While this is briefly touched, I would recommend providing a supplementary table with the reasons for the blood draws. Also, a comparison of the study population with the overall population would be helpful, e.g. smoking status of the parents, vaccination rates, day-care facility, health-care workers and highest education.

Clarity of the regression model: The interpretation of the regression model is rather difficult. Firstly, I am not sure how the first variables (top of table 3) were included in the model: was there an interaction term between age and infection risk? Secondly, I understand that table 3 shows the results of two regression models: (a) with outcome OI and (b) with outcome SI, and all variables shown went into the same model, respectively? Thirdly, I would argue that the association between infection risk and OI serves as a validation of your study approach, rather than being an actual research finding. Also, there is supposedly a strong interaction between the two explanatory variables infection risk and variant, which may be the reason for the missing significant effect in the youngest age group.

Modelling approach: I am not convinced that the stepwise inclusion approach was the most reasonable, as it is usually preferable to select variables based on theoretical background/ a causal diagram rather than on likelihood ratio tests. I would suggest selecting a number of variables based on theoretical background and keep all of them in the model irrespective of whether they are significant at a certain alpha-level. However, please make sure to avoid autocorrelations. I assume the authors had specific reasons to choose the model building approach. If you insist on this approach, would you please clarify your reasons?

Style: At several places throughout the manuscript, the language is not very precise or the content is not to the point, which partly makes it difficult to understand and lengthy to read. I suggest going through the text more closely in order to find such places. One special example is the conclusion: “Additionally, we saw higher rates during Delta and Omicron variants were dominant. Overall, the SI rate in undiagnosed and unvaccinated children and adolescents was 5.3%.” I don’t understand the first sentence, and the second sentence doesn’t really seem to be an adequate last sentence for a conclusion. Another example is in line 129: “The cumulative number of infections was normalized by the number of children in each age group of the three districts to adjust for the risk of being infected so far.” I don’t understand what was done on which numbers for what purpose.

Importance: the importance and urgency of this study are exaggerated. While it is important to describe the value that the study adds to the current knowledge, some phrasing is a bit much, e.g. line 72 “Data for infection- and vaccination-derived seropositivity among children and adolescents are urgently needed as seroepidemiologal data for pediatric population are sparse in this region.” Why would this be urgent?

Discussion: The discussion of seroprevalence rates and SI rates presented in other studies is lengthy and doesn’t provide much structured information. The authors should rather focus on the messages that they would like the reader to understand and use literature to support this than provide “random” findings from other studies.

SI: Throughout your manuscript, SI is treated as something undesirable. I understand that asymptomatically infected persons may pose a risk to close contacts. However, it is generally not a bad thing to acquire seropositivity without experiencing (severe) symptoms. Maybe it could help to discuss SI from different angles.

Table 1: (a) What is “vaccination availability”? (b) Vaccination and infection variable: were there any vaccinated AND infected children (Figure 4 suggests this)? (c) Day-care facility: All children <5 were in nursery or daycare? Some of the younger ones must have been taken care of at home. (d) smoking parent almost 50%. This seems very high.

Line 59: “The pandemic is characterized…” The pandemic has officially ended, so I suggest using past tense.

Line 118 “Overall infection (OI) was defined as seropositive tested samples and anamnestically infected participants.” Irrespective of negative test results?

Line 306: “The revised test strategy in schools may have led to rapid detection of infections and subsequent early block of infection chains in the families.” This sentence doesn’t really make sense to me, as you looked at school children, not their families.

Line 327: “Interestingly, job insecurity of the parents turned out to be negatively associated with OI in children. This may reflect stricter adherence to hygiene measures and restrictions in these families but needs to be further investigated.” Or maybe more likely, this could be a chance finding, which is likely given the multitude of different models that you tested. I suggest being more cautious with the interpretation of estimates.

Reviewer 2 ·

Basic reporting

1. Introduction part: Why is there limited data on SARS-CoV-2 seroprevalence in rural areas and an urgent need for data for seropositivity among children and adolescents in these regions? In other words, what research gap would this study fill? Provide more context on why this study is important.
2. The language needs polishing for better readability and grammatical correctness. For example, "evaluate of seroprevalence" should be "evaluate the seroprevalence."(Line 33); (Line 294-295) Maier et al. aiming to estimate the rate of fully susceptible children in the 16 federal states of Germany by analyzing official data from “SurvStat” for establishment of an infectious model; Line 318 needs a period, etc.
3. Correct this:(Line 89-90) Three patients with known immunodeficiencies 90 or administration of immunoglobulins were excluded. There were no other exclusion criteria.
4. The ORs and p values of factors that were not associated with SI and OI would be better presented in table 3.
5. (Line 152-153): Sex distribution across the three age groups was slightly, but significantly different (female: 46.7%, 46.6% and 55.0%, p=0.02). It is unclear what the three age groups were.
6. Some sentences (e.g. Line 166-167, Line 175, Line 189) are redundant in the result section since they were methods. Fig 3 (Line 191) should be Fig 2b.
7. In the result section, specified numbers should be presented along with the description. For example: lines 192-194, 196-197, etc.
8. Strengths and limitations would be better moved to the front of the conclusion.
9. Line 317: What “other groups” means?

Experimental design

1. Did the authors perform the sample size calculation? If not, why?
2. Could authors conduct pairwise comparisons of three groups?
3. Did the authors perform the missing data imputation in the complete case analysis (Line 125)?
4. Have the authors considered the 5 to 10 positive events per variable in the logistic regression?

Validity of the findings

1. Please add a flow chart of the study.
2. The statement of “Overall, the pediatric population of the study region is well represented, and the observed participant numbers follow an ideal representative distribution for all three districts with the exception of a slight over-representation in 1-year old children.”(Lines 154-156) should be in discussion part, supported with more detail, explanation the reason of “a slight over-representation in 1-year old children” and discussion of its potential impact on the study’s findings.
3. Could the authors give an in-depth analysis of how specific governmental measures and school testing strategies impacted SI rates? This could provide new insights into effective public health interventions.
3. (Line 169-170) The data presented is different from that shown in Figure 2a.
4. Line 210: Supplemental Figure S3 is the regional daily infections in children.
5. Some content should be moved from the result to the discussion. For example: Lines 223-226, 229-232.

Additional comments

no comments.

Reviewer 3 ·

Basic reporting

The study addresses a significant gap in understanding SARS-CoV-2 seroprevalence in rural northern Germany, focusing on a critical demographic—children and adolescents, from December 2020 to August 2022. In this cross-sectional study, 1166 blood samples from participants aged 6 months to 17 years were analyzed for SARS-CoV-2 specific antibodies. The findings suggest that government measures and systematic testing in schools likely contributed to the low silent infection rate observed. The methodology is sound, and the findings are significant for informing public health strategies. Addressing the identified weaknesses could further strengthen the manuscript.

Background:
Please elaborate on the seroprevalence rate and silent infection rate among children in urban regions of Germany, along with the corresponding sampling approach (line 69-70). In addition, although seroprevalence data is unavailable for the Pomerania region, was there any prevalence estimation using other methods (PCR and/or at-home antigen tests)? Please also report the pediatric vaccination rate during the study period. Such data could provide readers with a comprehensive background overview.

Similarly, please provide a brief summary of the risk factors identified in previous studies conducted on children in both urban and rural areas of Germany.

Experimental design

Materials and Methods:
Repeated participants should be adjusted for in the model (line 91-92). While repeated participation is beneficial in terms of sample size, it may also introduce issues with correlated data. The methods should describe how this was accounted for in the analysis, such as using statistical techniques to handle repeated measures.

From Supplementary Method S1 and manuscript lines 105-109, it is still unclear how the validity of estimating seroprevalence for the corresponding districts using the study cohort estimations was justified. The cohort participants received care from healthcare facilities and are probably more vulnerable to SARS-CoV-2 infections, suggesting they may not be representative of the underlying child population.

Statistical analysis line 123: Fisher’s exact test is typically used for 2x2 contingency tables. Since there are three age groups, the power of the test might be reduced, and adjustments for multiple comparisons should be considered (e.g., Bonferroni correction).

The multivariable regression model (line: 123-128) lacks robustness due to the complete cases method for handling missing data and the arbitrary nature of stepwise methods. Several sensitivity analyses are recommended to enhance the robustness of the findings: conducting multiple imputation to handle missing data more effectively, and considering methods like cross-validation, bootstrapping samples, and LASSO for more robust model selection.

Please specify the threshold used for VIF to decide on the presence of multicollinearity (line 132-133).

Validity of the findings

Results:
Line 208-220, please provide 95% CI when reporting Odds Ratios.

Discussion:
Limitation of Selection Bias Due to Hospital-Based Recruitment (line 260-261): while comparisons to other studies are helpful, this limitation could be strengthened by discussing specific characteristics of the study population that might differ from the general population (e.g., over-representation of 1-year-old children). Also, explaining any steps taken to mitigate selection bias in future studies would be useful.

Limitation of Missed Confounding Variables (lines 262-264): please provide examples of potential confounders that were not accounted for and discuss the potential impact on the study’s findings. Suggesting methods for improving confounder control in future studies would also be beneficial.

---

## Round 0.2 · Minor Revisions

Please address the remaining minor issues identified by one of the reviewers.

·

Basic reporting

Thank you for thoroughly revising your manuscript, especially regarding the methods and discussion! I still have several minor comments that you should address:


Methods: I would say the first paragraph under “exposure” fits better under the heading “outcome”.

Methods: I have read the methods, the supplementary methods and the footnote of table 3 and I am still unsure if I understand the variable “infection risk” correctly. Did you use the cumulative weekly incidence numbers for the respective region and age group from the start of the pandemic until the time of inclusion into the study? Or did you only look at the preceding week? Or only at a specific period because seropositivity could have vanished after some months?
Could you please provide a more specific explanation for this variable? Maybe a graph or an example calculation could help with the understanding. Also, could you please add the rationale for the inclusion of this variable to the manuscript?

Table 1: Do you assume that the 3 smoking children under 5 are real or do you think this is a mistake in the data? Have you tried to verify this?

Supplementary Figure 2: I think the age definitions have been interchanged between table columns.

Discussion general: The discussion is much improved. However, it can still be improved concerning content and readability in general. Some specific suggestions are below:

Discussion, first paragraph: “Seroprevalence increased during the Delta and Omicron waves with vaccine approval and relaxation of hygiene measures. Household contacts with infected individuals posed highest risk for SI. This rate was remarkably low at the start of our study, ranging from 1.0% to 3.8%, and increased particularly with the rise of Omicron variants in an age-dependent manner from 2.6% to 28.0%.”
The age-dependent increase can easily be misunderstood. You mean that the SI at the start ranged between 1.0 and 3.8% and increased to up to 28.0% during the Omicron wave, right? I suggest including the term “SI” in the sentence. Also, I would say, there should not be specific numbers in the discussion, rather an interpretation/summary of the numbers.

Discussion: “During our analysis, we directly compared serological results before and after 6 January 2022, as we noticed that the results were sensible to the time of blood sampling. As NAbs and IgA-S1 show a remarkable decrease in IgG-S1 positive samples, this could be at least partly due to immune escape of the different SARS-CoV-2 variants.”
Do you mean “sensitive”? Could you rewrite these two sentences to be easier to understand? Here is a suggestion from AI: "During our analysis, we conducted a direct comparison of serological results obtained before and after January 6, 2022, recognizing that the timing of blood sampling influenced the outcomes. Notably, we observed a marked decrease in neutralizing antibodies (NAbs) and IgA-S1 in IgG-S1 positive samples. This decline may be attributed, at least in part, to the immune escape mechanisms of various SARS-CoV-2 variants."

Discussion – Strengths: I would omit this paragraph if the journal style doesn’t specifically ask for a strengths paragraph. Your age distribution point can go to the limitations section to discuss selection bias.

Discussion – Limitations: Maybe you could elaborate a bit more on the selection bias. If children in your sample are more vulnerable than the general population, what could that mean for your findings? How would you expect the OI and the SI to differ from that of the general population? For example, I could imagine that the SI is lower in your sample than in the general population because parents worry more about the health of their children, which makes testing with only mild symptoms more likely.

Conclusion: This sentence “In the study sample, the rate of SI in undiagnosed and unvaccinated children and adolescents was 5.4%.” doesn’t seem to fit for the conclusion. This is a result, not a conclusion.

Experimental design

no comment/see basic reporting comments

Validity of the findings

no comment/see basic reporting comments

Additional comments

no comment/see basic reporting comments

Reviewer 2 ·

Basic reporting

no comments

Experimental design

Since the authors have performed the sample size calculation, the process would be better presented in the article or the supplemental materials.

Validity of the findings

no comments

Additional comments

no comments

Reviewer 3 ·

Basic reporting

I would like to commend the authors for their thorough and thoughtful revisions. The changes made to the manuscript have satisfactorily addressed the concerns I previously raised. The clarifications provided have strengthened the overall quality and rigor of the study. Given these improvements, I believe the manuscript is now in an excellent form and is suitable for publication in its current state.

Experimental design

NA

Validity of the findings

NA

Additional comments

NA

---

## Round 0.3 · accepted · Accept

Thanks for addressing all of the reviewers comments. The manuscript is ready for publication.